# Palladium-Doped Single-Walled Carbon Nanotubes as a New Adsorbent for Detecting and Trapping Volatile Organic Compounds: A First Principle Study

**DOI:** 10.3390/nano12152572

**Published:** 2022-07-27

**Authors:** Mehdi Yoosefian, Elaheh Ayoubi, Leonard Ionut Atanase

**Affiliations:** 1Department of Chemistry, Graduate University of Advanced Technology, Kerman 76311, Iran; 2Department of Nanotechnology, Graduate University of Advanced Technology, Kerman 76311, Iran; elaheayoobi@yahoo.com; 3Faculty of Medical Dentistry, Apollonia University of Iasi, 700511 Iasi, Romania; 4Academy of Romanian Scientists, 050803 Bucharest, Romania

**Keywords:** single-walled carbon nanotube, palladium, adsorption, acetonitrile, styrene, perchloroethylene

## Abstract

Volatile organic compounds (VOCs) are in the vapor state in the atmosphere and are considered pollutants. Density functional theory (DFT) calculations with the wb97xd exchange correlation functional and the 6-311+G(d,p) basis set are carried out to explore the potential possibility of palladium-doped single-walled carbon nanotubes (Pd/SWCNT-V), serving as the resource for detecting and/or adsorbing acetonitrile (ACN), styrene (STY), and perchloroethylene (PCE) molecules as VOCs. The suggested adsorbent in this study is discussed with structural parameters, frontier molecular orbital theory, molecular electrical potential surfaces (MEPSs), natural bond orbital (NBO) analyses, and the density of states. Furthermore, following the Bader theory of atoms in molecules (AIM), the topological properties of the electron density contributions for intermolecular interactions are analyzed. The obtained results show efficient VOC loading via a strong chemisorption process with a mean adsorption energy of −0.94, −1.27, and −0.54 eV for ACN, STY, and PCE, respectively. Our results show that the Pd/SWCNT-V can be considered a good candidate for VOC removal from the environment.

## 1. Introduction

In addition to the rapid growth of the world’s population, industrialization, unplanned urbanization, agricultural activities, and the excessive use of chemicals have contributed to environmental pollution. Today, the release of chemicals into the environment is a serious problem that affects both water and air quality. Volatile organic compounds, or VOCs, are compounds that range from hydrocarbons and halocarbons to oxygenates [1]. These compounds are organic chemical compounds that have a high vapor pressure at room temperature. VOCs are in the vapor state in the atmosphere and are considered pollutants [2,3]. Volatile organic compounds play a key role in the formation of secondary pollutants such as ozone through chemical reactions in sunlight with nitrogen oxides [4,5]. Trees and plants are natural sources of VOC emissions and, from the point of view of ozone formation, emit phosgene, which is one of the most reactive hydrocarbons [6,7]. However, the largest sources of VOC pollutants are gasoline vehicles [8,9]. Solvents used in paints and other household appliances and industrial products are other sources of VOCs [10]. Some of these substances are harmful in themselves, including benzene, which can cause cancer [11]. On the other hand, some of them can react with other gases and pollutants in the air and create harmful substances again [12]. Many VOCs are known to be hazardous air pollutants and are emitted by various industries [13]. Based on the preliminary results of studies from 2007, vehicle manufacturing (vehicles), furniture manufacturing (furniture), printing, bio-pharming, electronic manufacturing (electronics), and equipment coating (equipment) were selected as typical industries of VOC emission [14]. VOCs have the potential to affect human health and the environment at regional and global levels. Exposure to volatile organic matter can have various health effects on the central nervous system, hematopoietic system, chromosomal mutations, and other internal organs such as the liver and kidneys [15]. The special effects of these compounds include their role in reducing stratospheric ozone, global warming, and toxic and carcinogenic effects on human health [16,17]. So far, many studies have been performed to measure VOCs in the air or their urinary metabolites [18,19,20].

Acetonitrile (ACN) is a nitrile that is hydrogen cyanide in which the hydrogen has been replaced by a methyl group. It is an aliphatic nitrile and a volatile organic compound. ACN appears as a colorless limpid liquid with an aromatic odor. It is toxic and can be adsorbed through the skin. Its density is lower than water and its vapor is denser than air. ACN can cause poisoning in low doses [21]. Symptoms of intoxication include difficulty breathing, slow pulse, nausea, and vomiting. In severe cases, abnormalities and malnutrition can lead to death from respiratory failure. Cyanide ions induced by cyanide agents reversibly inhibit the activity of the enzyme cytochrome oxidase, which, despite the presence of oxygen in tissues, leads to suffocation and death [22,23]. Styrene (STY) is primarily a synthetic chemical. It is also known as vinylbenzene, ethenylbenzene, cinnamene, or phenylethylene. It is a colorless liquid that evaporates easily and has a sweet smell. STY is known to be carcinogenic, especially if it comes into contact with the eyes and skin, or is inhaled. Moreover, STY is mainly metabolized to styrene oxide in the human body [24], which is toxic, mutagenic, and carcinogenic. The US Environmental Protection Agency (EPA) has described styrene as a toxic substance to the gastrointestinal tract, kidneys, and respiratory system. Weakness, depression, hearing loss, and miscarriage have also been reported in women in the plastics industry (exposed to styrene) [25]. Perchloroethylene (PCE) is a colorless, highly toxic, and volatile solvent that has a wide range of applications in various industries, some of which may be limited by national or international standards. PCE is one of the chlorine solvents that has less toxicity than other halogenated derivatives. Properly used with safety equipment, the risk of toxicity will be negligible because it has good cleaning properties and high solubility [26]. One of the most common applications of solvents is the use of PCE in laundries. Exposure to this chemical has many negative effects and can affect various organs of the body, including the liver, kidneys, nerves, and heart. Among these chronic complications, neurological and psychological complications have a special place. These complications, which often have nonspecific and vague symptoms, are not easily detected. Symptoms of chronic encephalopathy include depression, memory loss, cognitive impairment, personality changes, psychosis, and decreased reaction time [27]. Other problems, such as decreased libido and hormonal changes, dementia, and Parkinson’s, have also been reported [28]. The removal of these pollutants from air or/and wastewater is necessary for many health and environmental considerations. To remove these pollutants, conventional methods such as reduction, precipitation, absorption, oxidation, and ion exchange have been used. However, the adsorption process was the most suitable method due to its high efficiency and economic considerations. Some adsorbents, such as activated carbon, zeol, biomaterials, nanoparticles, and polymers, etc., have been widely used for adsorption. However, the adsorption efficiency of these adsorbents is very low. Therefore, finding more efficient adsorbents has been the focus of various research groups.

Nanotechnology is a field of applied knowledge and technology that covers a wide range of applications [29,30,31]. Various nanostructure materials explored for VOC adsorption include activated carbons, fibers, films, nanoparticles, composite materials, and many kinds of carbon nanomaterial, where a major mode of gas entrapment is physical adsorption via van der Waals’ forces [32]. These include, for example, the adsorption of Benzene using ZnO nanoparticles coated on zeolite and activated carbons [33], chlorinated VOC using activated carbon [34], and VOC using TiO_2_–SiO_2_ films [35]. One of the most important nanostructures in terms of application and special properties is carbon nanomaterial [36]. Since the discovery of carbon nanotubes (CNTs), these nanotubes have become the preferred adsorbent among carbon nanomaterials due to their unique physical and chemical properties. CNTs are a new material that exhibit good adsorption behavior towards various toxic pollutants. These adsorbents have a fast adsorption rate and high adsorption efficiency. Furthermore, they are efficient in removing various pollutants and are easy to recover and reuse [37,38,39]. Because of the weak van der Waals interaction of the smooth nanotube surface with the adsorbents, the sensitivity and the selectivity of single-walled carbon nanotubes (SWCNTs) towards a specific compound can be improved by chemical functionalization [40]. One of the most effective modifications of the surface of SWCNTs, which introduces additional electronic states around the Fermi level and enhances their adsorption potential, is doping, i.e., heteroatom substitution into the lattice of SWCNTs [41]. Metal doping is frequently used to fabricate functionalized SWCNT sensors. Lu et al. [42] developed a Pd-functionalized SWCNT chemiresistor, which has a CH_4_ detection limit of up to 6 ppm. Adopting the same method, Oakley et al. [43] provided an H_2_ sensor with a detection limit of up to 10 ppm. Yoosefian [44] investigated the chemical and electronic properties of N_2_O adsorption on intrinsic CNT and Pd-doped CNT. Pd-CNT exhibits excellent adsorption properties towards gas molecules and increases N_2_O adsorption energies because of a strong affinity. Yoosefian argued that intrinsic CNT cannot serve as an effective adsorbent, while Pd-CNT can be designed as a novel carbon nanotube gas sensor with high sensitivity, fast response, and high efficiency, which can be used to detect N_2_O gas as an air pollutant. One of the methods used to scientifically study phenomena is the computational method, in which scientists seek to discover the logical relationships between phenomena and the features they observe. In this study, we propose a theoretical computation of Pd-SWCNTs to systematically identify and eliminate the ACN, STY, and PCE as volatile organic compounds using density functional theory (DFT). 

## 2. Materials and Methods

The optimization of molecular structures was calculated using the DFT method at the level of WB97XD (a range-separated version of Becke’s 97 functional with additional dispersion correction) and the standard basis set 6-311+G(d,p) without any symmetric considerations. All calculations were performed using the Gaussian 16 program [45]. The absence of imaginary frequency at the same level verified the optimized structures correspond to the energy minima. After full structure relaxation, optimized geometries of all investigated molecules in this study are depicted in Figure 1. (5, 5), an armchair SWCNT with 130 carbon and 20 hydrogen atoms (C_130_H_20_) were selected. A vacancy was made by removing one carbon atom in the sidewall of a SWCNT and replacing it with Pd to construct Pd/SWCNT-V. The diameter of the Pd/SWCNT-V system was 7.5 Å and its length was 16.6 Å. The average bond length of C–C and Pd-C were 1.43 and 1.99 Å, respectively. Adsorption of ACN, STY, and PCE molecules onto the Pd/SWCNT-V system was investigated subsequently. Adsorption energies were calculated according to Equation (1):(1)$$E_{ads}=E_{complex}-(E_{VOC}+E_{Pd/SWCNT-V})$$
where $E_{complex}$ is the total energy of the Pd/SWCNT with the VOC molecule and $E_{Pd/CNT-V}$ and $E_{VOC}$ is the total energy of the Pd/SWCNT-V and VOC molecules in relaxed geometry, respectively. The noncovalent interactions of the VOC molecule with the Pd/SWCNT-V were considered via different initial configuration complexes through the perpendicular direction to the SWCNT to reduce the unfavorable interactions.

DFT-based chemical reactivity and stability descriptors, which are electronic chemical potential (μ) and chemical hardness (η), were calculated as defined in Equations (2) and (3) [46,47].
(2)$$\mu ={\left(\frac{\delta E}{\delta N}\right)}_{V\left(r\right)}≅\frac{\varepsilon_{LUMO}+\varepsilon_{HOMO}}{2}$$
(3)$$ɳ={\left(\frac{\delta \mu }{\delta N}\right)}_{V\left(r\right)}=1/2{\left(\frac{\delta^{2}E}{\delta^{2}N}\right)}_{V\left(r\right)}$$

The natural bond orbital (NBO) [48] was calculated to quantify the charge transfer between the VOC and the Pd/SWCNT-V at the WB97XD/6-311+G(d,p) level. The AIM 2000 package was employed to achieve deep understanding of the nature of interactions in different investigated complexes via the Bader quantum theory of atoms in molecules (QTAIM) [49]. The sensitivity of Pd/SWCNT-V system as a new adsorbent for VOC was further analyzed by the calculation of density of stats (DOS).

## 3. Results and Discussion

### 3.1. Geometry and Adsorption Energy

A central C atom in the SWCNT is substituted by a Pd atom, forming a pyridine-like structure. The nearby hexagonal rings in the doping region deform through doping of the large Pd atom. In Figure 1 panel (I), a Pd-loaded SWCNT, Pd-C distances are 2.047, 1.97, and 1.97 Å and the Pd atom protrudes outside of the SWCNT surface and the geometry of the tube is changed due to the strong Pd-SWCNT interaction. So, the Pd-loaded SWCNT will be stable enough towards environmental effects due to the strong bond of the Pd atom to the nanotube. 

Figure 2 shows the fully optimized structure of the VOC compounds studied in this paper, which are adsorbed on the surface of the Pd/SWCNT-V. In this figure, the situation of VOC compounds with the least unfavorable interactions from the side and top view is shown.

To evaluate the adsorption strength, adsorption energy as one of the most important parameters was investigated. The calculated values of the adsorption energies (Eads), the total energy of compounds (Etot), and the dipole moments of investigated systems are given in Table 1.

The adsorption energies are −0.94, −1.27, and −0.54 eV for the ACN-Pd/SWCNT-V, STY-Pd/SWCNT-V, and PCE-Pd/SWCNT-V systems, respectively. These highly exothermic adsorption energies predict the strong interaction between Pd and VOC compounds through a chemisorption process. When the VOC compounds are adsorbed on the Pd/SWCNT-V, some changes occur in the structures of both compounds, especially in the adsorption site. The most important geometrical parameters, such as the bond length and bond angles containing the percentage of changes between initial adsorption position and the adsorption equilibrium position, are reported in Table 2. 

According to these results, for the ACN-Pd/SWCNT-V system, the largest change in bond length is related to the carbon−nitrogen triple bond in ACN, which increased by 1.528 percent to 1.156 Å. For the PCE-Pd/SWCNT-V system, after the adsorption of PCE, the bond length of the palladium atom with the surrounding carbons in the nanotube increases and the largest bond length change belongs to the carbon−carbon double bonds in PCE. Furthermore, in this system, a large change in the Cl–C–Cl bond angle was observed, i.e., it changed from 120° to 110.797°. Finally, for the STY-Pd/SWCNT-V system, the bond lengths of the STY molecule in the benzene ring decreased, and the rest of the bond lengths of the STY molecule increased after adsorption. The largest bond length change belongs to the Pd−C bonds in the Pd/SWCNT-V. Among the changes in the bond angles before and after STY adsorption onto Pd/SWCNT-V, we can mention the change in the C=C–C bond angle from 120° to 126.665°.

All of these geometrical changes after adsorption of VOC molecules and obtained adsorption energy values in this section confirm that the Pd/SWCNT-V system is an efficient and effective system for the adsorption and removal of VOC molecules. 

### 3.2. AIM and NBO Analyses

The theory of atoms in molecules (AIM) is applied to investigate the relationship between geometrical and topological parameters. Topological features of electron density ρ(r) have been associated with bond critical points (BCPs) [39]. The line of maximum charge density that links the nuclei is called a bond path and the (3,−1) BCP is referred to as a bond critical point. A (3,−1) BCP has the property of accumulating electron density at the BCPs in the perpendicular plane of the internuclear axis between a pair of nuclei which are considered to be bonded between them. The calculated ρ(r) and its Laplacian ∇^2^ρ(r_c_) show the nature of molecular bonds [50]. The properties of the Laplacian of the electron density, which was first used by Koch and Popelier, determine where the electron density is locally concentrated, ∇^2^ρ(r_c_) < 0, and where the electron density is locally depleted, ∇^2^ρ(r_c_) > 0. Figure 3 presents a typical molecular graph obtained by AIM analysis for the studied complexes. Small red spheres represent BCPs. As shown in this figure, the four BCPs that approve the interaction between the VOC and the Pd/SWCNT-V are specified. 

The most important topological parameters and the local electronic potential energy V(r_c_) [51] for the ACN-Pd/SWCNT-V, STY-Pd/SWCNT-V, and PCE-Pd/SWCNT-V systems are listed in Table 3.

The results reported in Table 3 show a good interaction between VOC molecules and the Pd/SWCNT-V. AIM analysis shows that a BCP between the Pd and VOC molecules is present in all investigated systems. The small value of ρ(r) and positive value of ∇^2^ρ(r_c_) for ACN-Pd/SWCNT-V, STY-Pd/SWCNT-V, and PCE-Pd/SWCNT-V systems approve a closed-shell interaction. In this study, the good interaction indicates the high VOC adsorption ability of the Pd/SWCNT-V surface.

In the NBO analysis, the electronic wave functions are construed in terms of a set of non-Lewis and a set of occupied Lewis localized orbitals [38]. To clarify the charge transfer between the VOC and Pd/SWCNT-V, an NBO calculation was carried out. The NBO method was also applied here to evaluate the second-order perturbation stabilization energy, *E*^(2)^. Clearly, the larger the *E*^(2)^ value, the more intensive the interaction between the VOC molecules and the Pd/SWCNT-V. Some of the most important large charge transfers that have been involved in the adsorption procedure are reported as follows (threshold for reporting: 0.344 eV):

For the ACN-Pd/SWCNT-V system: LP*(5)Pd to N≡C (*E*^(2)^ = 0.258 eV); LP(1)N to LP*(6)Pd (*E*^(2)^ = 2.608 eV); and LP(1)N to LP*(5)Pd (*E*^(2)^ = 1.219 eV).

For the PCE-Pd/SWCNT-V system: C=C* to LP*(5)Pd (*E*^(2)^ = 0.397 eV and C–C to LP*(8)Pd (*E*^(2)^ = 0.350 eV).

For the STY-Pd/SWCNT-V system: LP*(5)Pd to C–C* (*E*^(2)^ = 0.386 eV).

The results of the NBO analysis show that in all the systems the lone pairs of Pd atoms can participate as donors and acceptors. The strongest intramolecular charge transfer interactions appear between LP(1)N of ACN and LP*(6)Pd in the ACN-Pd/SWCNT-V system. Acceptable charge transfers and the adsorption energies indicate the chemical adsorption process.

### 3.3. Molecular Orbital Analysis

The highest occupied molecular orbital (HOMO) and the lowest unoccupied molecular orbital (LUMO) are responsible for chemical reactions. HOMO determines the ability to give electrons and LUMO determines the ability to accept electrons, and the difference between these two orbitals is considered to be the chemical stability of molecules. A large HOMO-LUMO gap implies high kinetic stability and low chemical reactivity. HOMO-LUMO and their properties, such as their energy, are important parameters in quantum mechanics. Conceptual DFT-based reactivity and stability descriptors are calculated and presented in Table 4. 

Results show that the HOMO-LUMO gap for the Pd/SWCNT-V system increases after adsorption of the VOC on all the three complexes. For example, in the STY-Pd/SWCNT-V complex, this gap for the Pd/SWCNT-V system before and after adsorption of STY is found to be 4.55 eV and 4.94 eV, respectively. From Table 4, after the adsorption of VOCs on the Pd/SWCNT-V nanotubes, except the ACN-Pd/SWCNT-V system, the HOMO energy values became more negative, resulting in a higher HOMO-LUMO gap.

Molecular electrical potential surfaces (MEPSs) explain the three-dimensional charge distributions of complexes. MPESs allow us to imagine variable charged regions of the VOC-Pd/SWCNT-V. Information on the charge distributions can be used to determine how the VOC molecules interact with the Pd/SWCNT-V. It also shows the behavior of molecules and visualizes the shape and the size of molecules. To understand the electrostatic potential energy data, a color spectrum is used in which the blue and red colors present as the highest and lowest electrostatic potential energy values, respectively. All color spectrums could be found on the surface of the system which facilitates the electrophilic and nucleophile interactions of the VOC molecules with the Pd/SWCNT-V surface. The molecular electrostatic potential graphic representations for complexes are depicted in Figure 4. 

The positive regions (blue color; 0.35) are related to nucleophilic reactivity and the negative (red color; −0.01) regions of MEPS are related to electrophilic reactivity. It can be seen in the total electron density maps of the electronic densities that the VOC molecules strongly bonded to the surface of the Pd/SWCNT-V, as was previously predicted in terms of binding energy.

The considerable differences in frontier molecular orbitals are due to the large charge transfer, which affects the density of states close to the Fermi surface. The total DOS of the investigated systems were predicted to explain the electronic properties and the influence of the VOC molecules adsorbed on the Pd/SWCNT-V. In comparison with the Pd/SWCNT-V, the band gap of the VOC-Pd/SWCNT-V increased. A significant leftward move happens to the DOSs of the PCE-Pd/SWCNT-V around the Fermi level. These changes in the DOSs would result in a change in the conductance of the Pd/SWCNT-V nanotube (see Figure 5).

## 4. Conclusions

Pd-doped SWCNT have been recommended as a potential adsorbent for the detecting and/or adsorbing ACN, STY, and PCE molecules as VOC from the atmosphere. To discuss the adsorption process of VOCs on Pd/SWCNT-V density, the functional theory-based method has been applied. To predict the adsorption properties and mechanisms of STY-Pd/SWCNT-V, PCE-Pd/SWCNT-V, and ACN-Pd/SWCNT-V complexes, the geometrical and electronic structures, frontier molecular orbitals, NBO, AIM, DOS, and MEPS were analyzed. In the adsorption of VOCs, STY adsorption onto the Pd/SWCNT-V system is energetically favorable. Strong binding energies between VOCs and the Pd/SWCNT-V indicates the nature of the chemical adsorption. The obtained results demonstrate that the transition metal-loaded SWCNTs, as a super molecular ligand, could be able to adsorb VOCs. Results related to the binding energy, charge transfer, energy gap variation, and DOS redistribution showed that the proposed model for adsorbing VOCs can effectively adsorb all three investigated gases and can be considered a good candidate for VOC removal from the environment. Furthermore, the tendency to adsorb STY was greater than for other gases, and gas ACN and PCE are placed in the next stage. Finally, it is important to point out that competitive conditions should be explored in further work to gain a more precise idea about the adsorption properties of the VOC.

## Figures and Tables

**Figure 1 nanomaterials-12-02572-f001:**
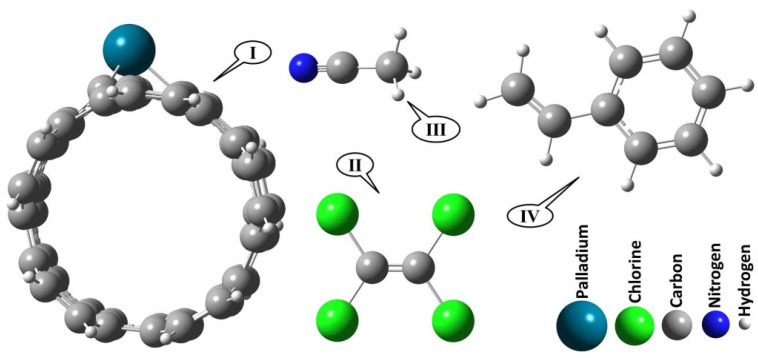
Optimized geometry of all investigated compounds in this study; (**I**) Pd/SWCNT-V, (**II**) PCE, (**III**) ACN, and (**IV**) STY.

**Figure 2 nanomaterials-12-02572-f002:**
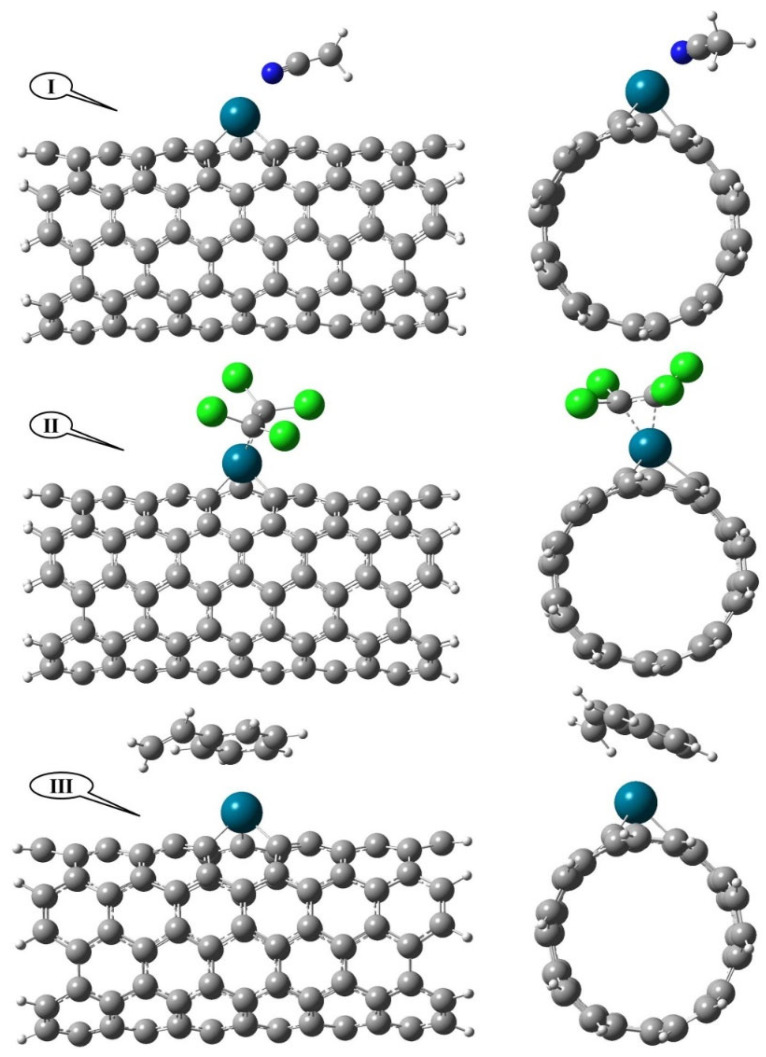
The optimized geometries of VOC-Pd/SWCNT-V complexes from side views and end views; (**I**) ACN-Pd/SWCNT-V, (**II**) PCE-Pd/SWCNT-V, and (**III**) STY-Pd/SWCNT-V.

**Figure 3 nanomaterials-12-02572-f003:**
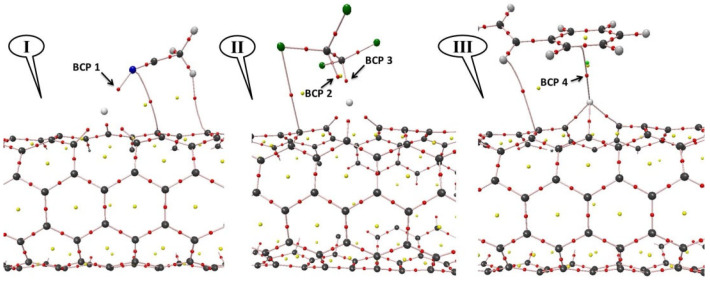
The molecular graphs for VOC-Pd/SWCNT-V complexes; (**I**) ACN-Pd/SWCNT-V, (**II**) PCE-Pd/SWCNT-V, and (**III**) STY-Pd/SWCNT-V.

**Figure 4 nanomaterials-12-02572-f004:**
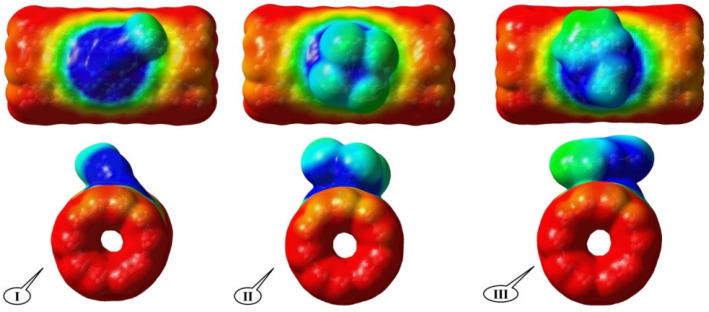
The electron density isosurfaces for VOC-Pd/SWCNT-V complexes from side views and end views; (**I**) ACN-Pd/SWCNT-V, (**II**) PCE-Pd/SWCNT-V, and (**III**) STY-Pd/SWCNT-V.

**Figure 5 nanomaterials-12-02572-f005:**
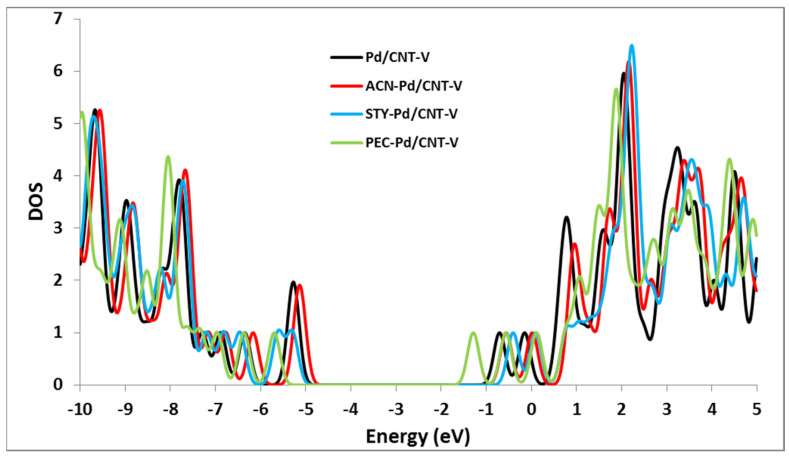
The Density of state diagrams for CAN-Pd/SWCNT-V, PCE-Pd/SWCNT-V, and STY-Pd/SWCNT-V compared to Pd/SWCNT-V.

**Table 1 nanomaterials-12-02572-t001:** Total energy (E_tot_ in eV), dipole moment (DP in Debye), and adsorption energy (Eads in eV) of the investigated compounds.

Compound	E_tot_	DP	Eads
ACN	−3568.42	3.88	-
STY	−8319.73	0.11	-
PCE	−51,799.97	0.00	-
Pd/SWCNT-V	−266,131.38	4.58	-
ACN-Pd/SWCNT-V	−269,700.73	9.07	−0.94
STY-Pd/SWCNT-V	−274,452.38	8.75	−1.27
PCE-Pd/SWCNT-V	−317,931.89	8.19	−0.54

**Table 2 nanomaterials-12-02572-t002:** Important geometrical parameters with percentage changes of investigated compounds before and after adsorption process (positive percentage changes mean an increase and negative values mean a decrease in geometrical parameters).

ACN	Before	After	% Changes	PCE	Before	After	% Changes	STY	Before	After	% Changes
Pd–C	1.968	1.972	0.188	Pd–C	1.968	1.981	0.652	Pd–C	1.968	1.970	0.105
Pd–C	2.047	2.033	−0.685	Pd–C	2.047	2.148	4.901	Pd–C	2.047	2.026	−1.045
Pd–C	1.968	1.972	0.190	Pd–C	1.968	2.021	2.706	Pd–C	1.968	2.076	5.499
N–C	1.139	1.156	1.528	C=C	1.355	1.410	4.053	C=C	1.355	1.324	−2.286
N–C–C	179.987	179.886	−0.056	C–Cl	1.760	1.822	3.516	C–C	1.540	1.484	−3.644
				Cl–C–Cl	120.000	110.797	−7.669	C=C–C	120.000	126.665	5.555

**Table 3 nanomaterials-12-02572-t003:** The topological parameters and the energy of the local electronic potential energy (V(rc)) for investigated compounds (all values in eV).

ACN	ρ (r)	∇^2^ρ (r_c_)	V (r_c_)	PCE	ρ (r)	∇^2^ρ (r_c_)	V (r_c_)	STY	ρ (r)	∇^2^ρ (r_c_)	V (r_c_)
Pd–C	3.424	4.159	−5.290	Pd–C	3.569	5.032	−5.715	Pd–C	3.169	3.025	−4.593
Pd–C	2.997	3.242	−4.227	Pd–C	3.305	8.204	−5.343	Pd–C	3.547	4.352	−5.604
Pd–C	3.724	5.046	−6.101	Pd–C	2.595	9.914	−3.938	Pd–C	3.552	4.352	−5.617
N–C (ACN)	12.526	−15.197	−41.648	C=C (PCE)	7.681	−21.102	−17.240	C=C (STY)	9.180	−29.096	−23.139
N(ACN)–Pd	1.564	8.878	−2.078	C(PCE)–Pd	2.636	7.559	−3.826	C–C (STY)	6.824	−18.822	−14.030
				C(PCE)–Pd	2.258	7.986	−3.133	C(STY)–Pd	0.721	2.132	−0.546

**Table 4 nanomaterials-12-02572-t004:** The energy of the HOMO (εH) and LUMO (εL); HOMO–LUMO gap (EG); chemical hardness (η); and chemical potential (μ) of the investigated compounds. All values are in eV.

Compound	εH	εL	EG	η	μ
Pd/SWCNT-V	−5.25	−0.70	4.55	2.28	−2.97
ACN-Pd/SWCNT-V	−5.11	−0.55	4.56	2.28	−2.83
PCE-Pd/SWCNT-V	−5.69	−1.13	4.57	2.28	−3.41
STY-Pd/SWCNT-V	−5.29	−0.35	4.94	2.47	−2.82

## Data Availability

The data presented in this study are available on request from the corresponding author.

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
