# Peer review of "Palladium-Doped Single-Walled Carbon Nanotubes as a New Adsorbent for Detecting and Trapping Volatile Organic Compounds: A First Principle Study"

_nanomaterials, 2022, doi:10.3390/nano12152572_

Round 1
Reviewer 1 Report
The author's report the theoretical analysis of palladium doped single-walled carbon nanotube as a new adsorbent for VOCs. The paper is well written. The following point should be clarified prior to publication.
1) Why is it that palladium is selected as doped element? How about other metals such as Pt, Au, etc?
2) The experimental data for adsorption test is not existed? I doubt that the present Pd-doped carbon nanotube can work as a adsorbent.
Reviewer 2 Report
The authors report a study of Pd-doped SWCNTs as potential adsorbents for the detection and/or adsorption of ACN, STY, and PCE molecules as environmental VOCs. To carry out this study, the authors used density functional theory to predict the adsorption properties and mechanism of the STY-Pd/CNT-V, PCE-Pd/CNT-V, and ACN-Pd/CNT-V systems in terms of geometric and electronic structures. The authors' results regarding binding energy, charge transfer, variation in energy gap, and DOS redistribution demonstrate that Pd/CNT-V is a potential candidate for the removal of VOCs from the environment.
From the reviewer's point of view, the authors' results could be interesting for the readers of the nanomaterials journal, but some issues should be addressed to improve the manuscript.
- The manuscript needs editing.
- The authors should discuss the influence of adsorbed VOCs on the adsorption properties of neighboring VOCs. The case of simultaneous adsorption on neighboring sites should be considered.
Reviewer 3 Report
Recommendation: Major revision
Comments
1) Once abbreviated, use the term VOC freely.
2) Page 1, Many VOCs are known to be hazardous…………………….. emitted from various industries. Give example of such industries here
3) Page no 2, the introduction about nanotechnology is lengthier and repeated. Please make it concise.
4) Before abbreviated, SWCNT should be appeared in the full form.
5) So far, various nanostructures such as fibers, films, nanoparticles, and composite materials have been used for VOC removal. The introduction should be upgraded with this information.
6) Since the authors have focused on MWCNTs, they must explain why MWCNTs are better than other materials for VOCs removal.
7) Role of palladium is totally missing in the introduction part.
8) Palladium is expensive material. Why did the author choose palladium for this study?
Reviewer 4 Report
THe paper is a theoretical study about the adsorption of some organic molecules on a material. THe fact that this is just a theoretical study has to be mentioned in the title, otherwise, it is confusing for the potential authors.
THe material that the authors are proposing is really expensive.. Pf and SWCNT... just for removal of VOCs. THere are a lot of procedures less expensive for removal of VOCs (other adsorbents, photocatalysis, or oxidation), and the introduction does not justify the use of these so expensive adsorbents. Introduction only focuses in describing the organic molecules and the fact that VOCs are pollutants, under my point of view, this information is not necesary for the especialized public of a peer review journal. So, the interest of this study is limited.
In addition to the poor interest, the validity of the study is limited since any experimental data is given in order to check the theoretical calculations, that depends of many factors. THe optimization of a supported material (Pd dispersed on a carbon material) is far from the simplicity presented in this paper, since in real circumstances it depends on the size of the Pd nanoparticle, the dispersion, the shape and the oxidation state.
I do not think that, in absence of experimental data, such a paper deserves publication,
Reviewer 5 Report
The manuscript entitled Palladium doped single-walled carbon nanotube as a new adsorbent for detecting and trapping volatile organic compounds submitted by the group of Authors represent an interestin theoretical study on the interaction an potential of suggested SWCNT-dopped woth Pd to detect and/or adsorb acetonitrile (ACN), styrene (STY) and perchloroethylene (PCE) molecules as VOCs.
Abstract should be brief and concise giving the reader a concentrated information on the topic.
introduction part should be focused on the topic, and not so general as in many cases trough the introduction part. Authors should introduce the readers to the topic that is the core of the work.
Could the Authors make more comment on the aromatic nature of suggested molecule STY and interaction woth SWCNT.
Round 2
Reviewer 2 Report
The authors have made an effort to improve the new version of their paper, and I suggest that they consider the comments below for the version to be published:
The chemical formulas should be written correctly, the numbers should be subscripted.
The reviewer thinks that it should be said at least in the conclusion, that may be to have a more precise idea about the adsorption properties of the gas, the properties in case of competitive conditions should be explored as well and that this will be the subject of further work.
Reviewer 3 Report
the paper can be accepted.
Author Response
Thanks!
Reviewer 4 Report
As I already mentioned with my first paper, I found this is a non-sense research approach with no interest for academics. THe authors reponses are vague and do not address my comments already exposed in my first report. Subsequently, I cannot recommend the publication of this paper.
